# BioDisco: Multi-agent hypothesis generation with dual-mode evidence, iterative feedback and temporal evaluation

## Abstract

Identifying novel hypotheses is essential to scientific research, yet this process risks being overwhelmed by the sheer volume and complexity of available information. Existing automated methods often struggle to generate novel and evidence-grounded hypotheses, lack robust iterative refinement and rarely undergo rigorous temporal evaluation for future discovery potential. To address this, we propose BioDisco, a multi-agent framework that draws upon language model-based reasoning and a dual-mode evidence system (biomedical knowledge graphs and automated literature retrieval) for grounded novelty, integrates an internal scoring and feedback loop for iterative refinement, and validates performance through pioneering temporal and human evaluations and a Bradley–Terry paired comparison model to provide statistically-grounded assessment. Our evaluations demonstrate superior novelty and significance over ablated configurations and generalist biomedical agents. Designed for flexibility and modularity, BioDisco allows seamless integration of custom language models or knowledge graphs, and can be run with just a few lines of code.

## 1 Introduction

Advancing biomedical research depends on a supply of novel, testable hypotheses. However, generating such hypotheses traditionally relies on human intuition, domain expertise and manual literature reviews, all of which are increasingly strained by an exponential growth in scientific data and publications. This challenge motivates automated approaches to scientific discovery that can systematically uncover and integrate complex biological relationships. Moreover, robust evaluation of such tools is essential.

### 1.1 Background

Knowledge graphs (KGs) provide a structured foundation for storing and analyzing scientific knowledge, with biomedical KGs capturing complex relationships among genes, proteins, diseases, drugs and biological pathways: e.g. HetioNet (Himmelstein et al., 2017) and PharmKG (Zheng et al., 2020). While a variety of tools have been developed to mine such graphs for novel insights (Perdomo-Quinteiro & Belmonte-Hernández, 2024), construction and maintenance of KGs are time-consuming and labour-intensive tasks, and a significant proportion of biomedical knowledge remains untapped in unstructured text.

Large language models (LLMs) have emerged as a promising solution, capable of extracting relevant information and identifying latent patterns from vast corpora (Brown et al., 2020), with greater capacity for contextual reasoning than earlier methods of hypothesis generation (c.f. Spangler et al., 2014). However, LLMs risk returning plausible but factually groundless information (Bélisle-Pipon, 2024). To mitigate this, models can be imbued with external interfaces, enabling dynamic literature retrieval via retrieval-augmented generation (RAG; Lewis et al., 2020; Singhal et al., 2023).

Evaluating the output of these generative systems—particularly for novel hypothesis generation, in absence of 'ground truth'—presents its own significant challenge. Whilst general evaluation strategies exist, including expert-based assessments, automated semantic similarity scoring and validation against curated benchmarks (Liu et al., 2025), many recent approaches rely heavily on model-based evaluation, i.e. using other LLMs as

judges (Li et al., 2024). Though more accessible than consulting human experts, model-based evaluation is sensitive to the relative strengths of the generator and the judge (Dorner et al., 2024) and alone may not fully capture a system's capacity for genuine scientific discovery.

## 1.2 Related work

### 1.2.1 Multi-agent systems

Recent research has explored the use of 'specialized agents', often within multi-agent systems, to model the iterative and collaborative nature of scientific discovery. These approaches typically leverage LLMs augmented with tools, assigning distinct roles to individual agents to simulate the research process (Ren et al., 2025). Examples include AI Scientist (Lu et al., 2024), which automates the full research pipeline, ResearchAgent (Baek et al., 2025), a multi-agent system using KGs and peer review-style feedback, and Google's Co-scientist (Gottweis et al., 2025), which employs a generate–debate–evolve loop. In the biomedical domain, AI agents like Biomni (Huang et al., 2025b) offer generalist capabilities from literature curation to experimental design. More narrowly focussed systems include SciAgents (Ghafarollahi & Buehler, 2024) for biomaterials discovery and IntelliScope (Aamer et al., 2025) for biomedical hypothesis generation by identifying meaningful paths in KGs.

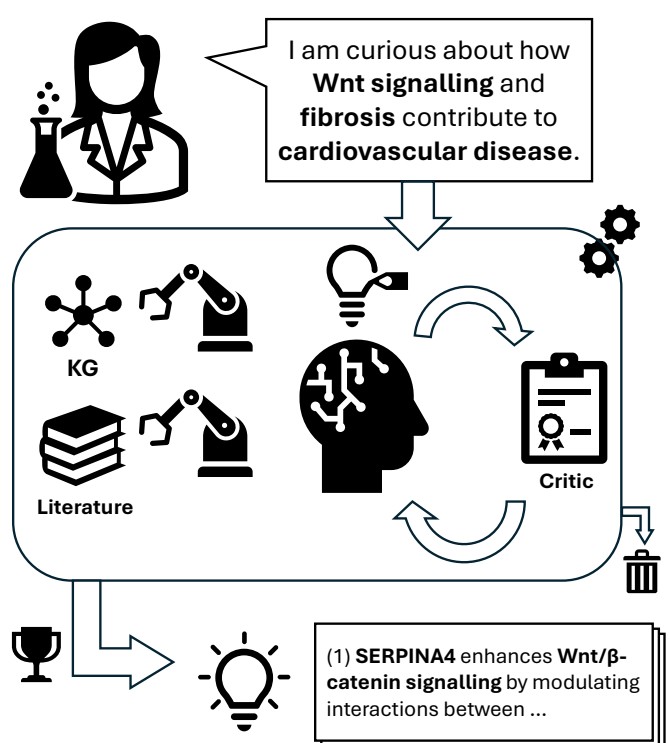

Figure 1: A high level overview of our automated framework for hypothesis generation. Overseen by a planner, agents search academic literature and query a knowledge graph to obtain articles and subgraphs relevant to a user-specified research topic. A scientist agent integrates these sources to derive initial hypotheses, which are rated by a critic, then refined with additional background, discarded or presented to the user with supporting evidence

### 1.2.2 Evaluation metrics

Evaluating the efficacy of automated hypothesis generation systems, particularly at scale, involves assessing not only coherence but also novelty, experimental feasibility and potential impact. A prevalent method involves testing a system's ability to rediscover known scientific findings using historical data (Sybrandt et al., 2018; 2020), though this 'temporal evaluation' on held-out data remains underutilized. Assessment by human domain experts is desirable but challenging to perform reliably for large-scale evaluations (Alkan et al., 2025), thus Qi et al. (2024) used LLMs as judges, rating hypotheses for novelty, verifiability, significance, and relevance. This and other LLM evaluations rely on assigning numerical ratings to individual outputs. However, Liusie et al. (2023) showed that paired comparisons using LLMs can outperform direct scoring approaches, at the risk of introducing additional sources of bias, such as order effects. Some authors, e.g. Gottweis et al. (2025), used Elo ratings for automated evaluation. However, a more principled statistical approach would use a probabilistic model to provide uncertainty estimates for ratings and enabling model diagnostics; the Elo system (1978) provides neither.

### 1.2.3 Benchmarks

Recent efforts further underscore the need for standardized evaluation frameworks. Benchmarks such as PubMedQA (Jin et al., 2019), GPQA (Rein et al., 2024), and CARDBiomedBench (Bianchi et al., 2025) provide question–answer-based datasets to assess LLM reasoning, with a focus on the biomedical domain. Similarly, TruthHypo (Xiong et al., 2025) employs a temporal split to benchmark the classification of associations among biological entities. Complementary initiatives, including HypoBench (Liu et al., 2025), DiscoveryBench (Majumder et al., 2024) and LAB-Bench (Laurent et al., 2024), reinforce the growing emphasis on quantitative assessment of automated hypothesis generation systems. However, most of these benchmarks primarily assess knowledge retrieval, classification, or data-driven insights, falling short of evaluating the generation of novel, multi-entity textual hypotheses that our system aims to produce.

### 1.3 Contributions

We propose BioDisco, a novel multi-agent framework that uniquely integrates LLM reasoning with access to biomedical knowledge graphs and real-time scientific literature, featuring an internal multi-dimensional scoring mechanism for iterative hypothesis refinement. Our system orchestrates a pipeline of specialized agents that collaboratively construct, evaluate and refine biomedical hypotheses, consistently generating evidence-based insights and inferring verifiable relationships beyond its explicit knowledge base.

Unlike systems that treat hypothesis generation as a 'one-shot' task or lack deep domain knowledge integration, BioDisco employs a sophisticated self-critique loop grounded in dual-mode evidence. This paper presents the following:

1. BioDisco, a flexible and modular automated hypothesis generation framework, based on LLM agents augmented with interfaces to biomedical KGs and literature search;

2. a rigorous evaluation of the system's ability to predict unpublished scientific relationships, including:

   (a) a temporal evaluation on two held-out datasets of 'future' discoveries;
   (b) a model-based ablation study, leveraging a Bradley–Terry model accounting for ties and order effects;
   (c) a human evaluation by experts in two biomedical domains, modelled using item-response theory;

3. an open-source Python package of the BioDisco framework, available to install from PyPI.org via `pip`.

A detailed case study is also presented in the Appendix.

## 2 Multi-Agent Hypothesis Generator

The BioDisco framework, illustrated in Figure 1, operates through a sophisticated multi-agent architecture, where each agent has a distinct role to facilitate generation, evaluation, refinement and final selection of hypotheses. BioDisco is, to our knowledge, the first system to integrate KG querying, literature search, 'reviewer' agents and explicit numerical scoring within a unified agentic feedback loop, wherein each generated hypothesis is internally evaluated against multiple metrics and refined to enhance its quality. A comparison with other systems is presented in Table 1.

The core of BioDisco's functionality resides in its network of specialized agents, illustrated in Figure 2. The Background agent searches academic literature to gather scientific evidence, while the Explorer queries a biomedical KG to retrieve relevant subgraphs. The Scientist then formulates initial hypotheses by integrating information from the summarized literature and subgraph data, proposing novel associations. These candidate hypotheses are systematically evaluated by a Critic, who scores each one for novelty, verifiability, relevance and significance (following Qi et al., 2024), providing structured feedback. The Reviewer agent uses this feedback to formulate targeted refinement strategies, such as suggesting new entities or expanded literature review, and invokes dedicated interfaces to conduct more specialized KG queries or fine-grained

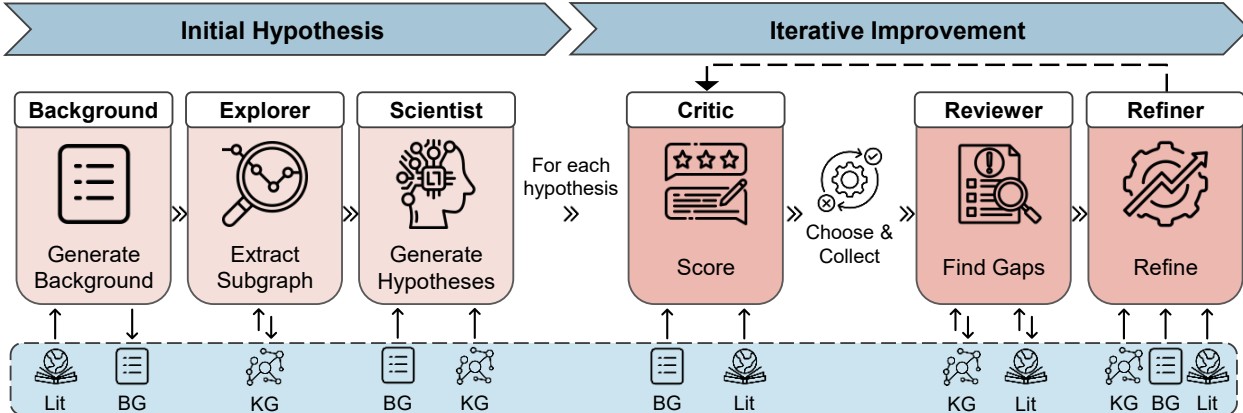

Figure 2: BioDisco architecture. Each agent has a distinct role, some augmented with external tools. Agents interact sequentially: the user's input is first processed by the BACKGROUND agent to generate a topical summary, guiding KG extraction by the EXPLORER and initial hypothesis generation by the SCIENTIST. Each hypothesis undergoes an iterative cycle of evidence retrieval and refinement. Finally, evaluations from the CRITIC agent are used to identify the most promising hypotheses. Here, Lit refers to the literature interface, KG to the KG interface, and BG to the generated background.

Table 1: Features of LLM agentic systems: multiple agents, use of external tools, knowledge graph integration, ability (dynamically) to search academic literature, presence of a 'reviewer' agent critically appraising outputs, and refinement of hypotheses based on explicit numerical scoring. ResearchAgent (Baek et al., 2025) constructs a knowledge graph from an academic graph, but does not access a 'live' search API

|  | Co-scientist | Intelliscope | SciAgents | ResearchAgent | BioDisco |
|---|---|---|---|---|---|
| Multi-agent | ✓ | ✓ | ✓ | ✓ | ✓ |
| Tool use |  |  | ✓ | ✓ | ✓ |
| Knowledge graph |  | ✓ | ✓ | ✓ | ✓ |
| Search |  |  | ✓ |  | ✓ |
| Reviewer |  | ✓ | ✓ | ✓ | ✓ |
| Scoring |  |  |  | ✓ | ✓ |

literature searches. The REFINER amends the hypotheses based on this guidance and additional information. The decision module monitors CRITIC scores to select and collect hypotheses throughout the feedback loop.

## 2.1 Dual-mode evidence grounding

To ensure the factual reliability of generated hypotheses, BioDisco exploits two complementary evidence sources: a structured knowledge graph and real-time scholarly literature retrieved via the PubMed API. These knowledge bases are dynamically queried throughout the hypothesis generation process via dedicated interfaces.

The **KG interface** enables dynamic retrieval of structured biomedical knowledge from an external graph. We adopt PrimeKG (Chandak et al., 2023) as the underlying KG, hosted in a Neo4j instance: queries are formulated using Cypher for efficient subgraph retrieval. For each query, agents supply a background summary and a set of keywords. These keywords are first mapped to canonical KG nodes via pretrained biomedical embeddings (`BioSimCSEBioLinkBERT-BASE`; Kanakarajan et al., 2022), then further filtered for contextual relevance. The resulting set of nodes defines a personalized query, which is executed to retrieve relevant evidence in the form of both direct and multi-hop relations. Query parameters, including relation types and traversal depth, are adaptively set based on task objectives and intermediate feedback. Returned subgraphs are summarized and converted into standardized plaintext representations for downstream use.

This approach optimizes retrieval for both efficiency and relevance, enabling flexible and context-aware access to large-scale structured knowledge.

The **literature interface** enables agents to programmatically access biomedical literature through an LLM-guided, context-aware query and retrieval workflow. At each invocation, agents supply a set of keywords. For background research, only keywords are provided, whereas for evaluation, the current hypothesis is included, and for revision, both the hypothesis and low-score feedback are appended. An LLM-based planning module analyzes these inputs to generate structured PubMed query strategies, grouping terms by semantic similarity and determining optimal Boolean logic for combining concepts. These strategies are converted into PubMed-API-compatible queries, including support for MeSH and TIAB fields as well as date-range constraints. If the initial strategy yields insufficient results, the interface automatically relaxes query constraints to increase recall.

## 2.2 Iterative refinement & multi-agent collaboration

The BioDisco framework employs a dynamic process of co-operative evaluation and improvement. This iterative self-critique loop, unique in its integration of an internal critic and (external) dual-mode evidence, is designed to enhance hypothesis quality and credibility of initially generated hypotheses. The preliminary generation stage is driven by user input and involves three specialized LLM agents. We demonstrate this with an illustrative example.

1. The user provides a research topic or set of keywords.

   "Role of GPR153 in vascular injury and disease"

2. The BACKGROUND agent searches biomedical literature to synthesize a textual summary of the research area.

   "GPR153 plays a crucial role in vascular injury responses by modulating critical signaling pathways such as cAMP, YAP/TAZ ..."

3. The EXPLORER queries the knowledge graph to obtain domain-relevant context, treating user-provided terms as seed entities and retrieving a local subgraph describing their connections.

   "Nodes: GPR153 (gene/protein), CEBPB ... Direct Edges: CEBPB → protein_protein → GPR153... MultiHop Paths: PHYHIP → ... TTR → molfunc_protein → PP2CA → ..."

4. Roleplaying a biomedical researcher, the SCIENTIST integrates the textual representation of the subgraph and the background summary to formulate initial hypotheses as novel natural language associations between two or more entities. Three such hypotheses are generated in parallel.

   "GPR153 activation in vascular smooth muscle cells enhances pro-inflammatory gene expression via the YAP/TAZ pathway ..."

For each initial hypothesis, an iterative refinement process is launched to progressively improve its quality and credibility. This critical loop, described in steps 5–7, cycles through evaluation, real-time information retrieval and targeted revision. This involves three additional agents:

5. The CRITIC evaluates each hypothesis alongside the retrieved background and literature summary, providing structured feedback in the form of numerical scores and comments detailing strengths and weaknesses.

   "Novelty: 4; Relevance: 5; Significance: 4; Verifiability: 4. The hypothesis can be tested using genetic manipulation ... however, the complexity of the regulatory networks may introduce challenges in isolating ... Overall Score: 17/20"

6. The REVIEWER agent identifies specific deficiencies (e.g. low novelty or weak evidence) and formulates targeted refinement strategies, such as suggesting new entities or an expanded literature search.

Different strategies include deeper knowledge graph queries via the EXPLORER, or refined literature searches via the BACKGROUND agent for additional evidence, and revisiting the original background to ensure alignment with the research topic.

> "Actions: Background, KG. Suggestions: Novelty - The hypothesis lacks exploration of the specific biological mechanisms ... Verifiability - The hypothesis lacks robust experimental support ..."

7. The REFINER modifies the candidate hypotheses based on the feedback and new evidence, adjusting phrasing, structure, or content as needed before returning the refined versions to the CRITIC.

> "GPR153 activation in vascular smooth muscle cells enhances pro-inflammatory gene expression by promoting CEBPB-mediated YAP1 signaling, thereby potentially integrating with EGR1 and GSK3B pathways to exacerbate neointima formation following vascular injury ..."

This feedback loop repeats for up to three cycles. After each round, the decision module checks whether the overall score exceeds a predefined threshold, allowing early exit for high-quality hypotheses or discarding consistently underperforming ones. Finally, the framework selects the highest-scoring and most scientifically valuable hypotheses as outputs with their scores and supporting evidence. Further technical details about individual agents and their system prompts are provided in the Appendix.

## 3 Evaluation of LLM-Generated Hypotheses

We designed a comprehensive, three-part empirical evaluation to assess the quality and novelty of hypotheses generated by BIODISCO.

Our temporal evaluation—described in the next subsection—assesses the system's capacity for genuine discovery, by determining if it is able to predict discoveries made after a certain time cutoff, with access only to data and literature before it.

This is followed by a model-based ablation study to quantify the contributions of the agentic framework, tools and refinement protocol. Significantly, our evaluation incorporates a pairwise LLM evaluator combined with a Bradley–Terry model. This statistical approach provides more stable and human-aligned assessments of hypothesis quality across multiple dimensions (Liusie et al., 2024), delivering not just point estimates but also uncertainty quantification via 95% comparison intervals, a significant advancement over methods that merely offer direct scores or unidimensional summaries.

Finally, a human evaluation records the expert judgment of 9 researchers in two biomedical subfields: cardiovascular disease and immunology. Their responses are modelled using a Bayesian polytomous Rasch model to account for possible rater-specific and metric-specific biases.

Except where otherwise specified, all agents were based on GPT-4.1 (knowledge cutoff: $1^{st}$ June, 2024).

### 3.1 Can BioDisco predict future hypotheses?

Firstly, we test if the system can generate hypotheses not present in its training corpora or knowledge interfaces.

### 3.1.1 Datasets

We used two publicly available datasets. (1) An unseen dataset from Qi et al. (2024), containing background–hypothesis pairs curated from biomedical literature, published in August 2023. We filtered this dataset to focus on core biomedical findings, removing entries with psychology or epidemiology backgrounds, resulting in 134 samples. (2) TruthHypo (Xiong et al., 2025) provides keyword pairs across three categories: chemical–gene, gene–gene, and gene–disease. Each pair is labelled with a relationship type: positive, negative, or no relation and was extracted from literature published after 2024. We focused exclusively on positive

Table 2: Performance of temporally-restricted BIODISCO on the TruthHypo benchmark for three categories of task

| Category | Precision | Recall | $F_1$ | Accuracy |
|----------|-----------|--------|-------|----------|
| Chemical–Gene | 0.90 | 0.90 | 0.90 | 0.90 |
| Disease–Gene | 0.82 | 0.82 | 0.82 | 0.82 |
| Gene–Gene | 0.83 | 0.83 | 0.83 | 0.83 |

and negative relations, excluding the 'no relation' category, as BIODISCO's primary focus is generation of plausible relationships between entities rather than confirming their absence. From this dataset, we created two subsets. We first selected 30 samples, balanced across categories, to configure a classifier agent. From the remainder, we sampled a test set of 300 instances (limited for computational feasibility), stratified across all categories and the two classes, for evaluation.

### 3.1.2 Setup

We enforced strict temporal splits by limiting BIODISCO's knowledge to information available up to December 2022 for the Qi et al. (2024) experiment, and up to December 2024 for the TruthHypo experiment. This was achieved by restricting access to PubMed articles published after these cut-off dates. We also used PrimeKG (published April 2022) as a knowledge source. To prevent knowledge leakage from the underlying LLM, we used the GPT-3.5 Turbo (knowledge cutoff: $1^{st}$ September 2021) for the Qi et al. (2024) experiment. To bridge BIODISCO's free-text generation with the structured labels required by the TruthHypo task, we developed a classifier agent to predict the relationship class from the textual hypotheses.

### 3.1.3 Results

We evaluated the semantic alignment between the generated and 'gold-standard' hypotheses from Qi et al. (2024) by computing their cosine similarity using a pre-trained biomedical sentence encoder, `BioBERT-mnli-snli-scinli-scitail-mednli-stsb` (Deka et al., 2022). As a baseline, we calculated pairwise similarities between unrelated 'gold' hypotheses (i.e. those associated with different background contexts), serving as a negative control for expected similarity in semantically unrelated pairs. Figure 3 shows that generated hypotheses were more semantically similar to the gold standard (median similarity: 0.68) than unrelated gold hypotheses were to each other (median: 0.34), demonstrating that BIODISCO reliably produces hypotheses closer to human-curated ground truth than expected by chance.

In the second experiment, we assessed whether BIO-DISCO could generate a hypothesis that correctly implies the relationship for a given entity pair. We used our classifier agent to perform binary classification (positive, negative) on the generated text.

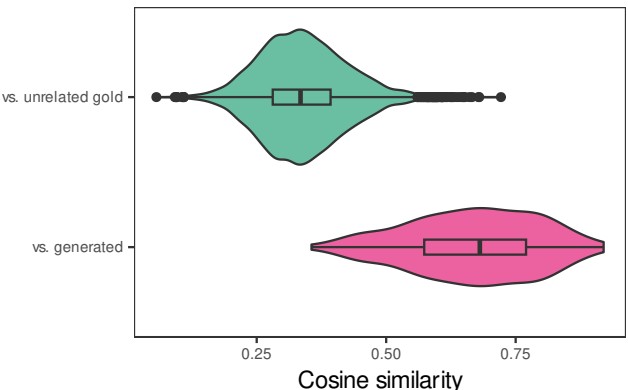

Figure 3: Violin plot demonstrating that hypotheses generated by BIODISCO are semantically more similar to 'gold' hypotheses than gold hypotheses are to other hypotheses. Top distribution shows pairwise similarity of unrelated gold hypotheses; bottom shows similarity of BIODISCO-generated hypotheses to gold standard for the same topics

Results are given in Table 2, showing the classifier achieved high precision and recall, indicating that the hypotheses generated by BIODISCO contain accurate and discernible relational signals.

These results indicate that BioDisco, operating under strict temporal constraints, can successfully infer novel, semantically relevant and factually verifiable hypotheses.

## 3.2 What are the effects of knowledge & refinement?

To evaluate the contribution of each component to the framework, we compared BioDisco against a series of ablated configurations. These included:

1. GPT-4.1 (baseline): a standalone single LLM, prompted without agentic structure or external tools.

2. Multi-agent system: a system mirroring our architecture, but without iterative refinement or knowledge interfaces.

3. Multi-agent + tools: a system with access to the literature and KG interfaces but without iterative refinement.

4. Multi-agent + refine: a system with an iterative refinement loop, but without access to knowledge interfaces.

5. BioDisco: the full system described in § 2.

For a direct comparison with other biomedical agentic systems, we also generated a series of hypotheses with Biomni (Huang et al., 2025b):

6. Biomni: a general-purpose biomedical AI agent, based on GPT-4.1, with access to various bioinformatics tools but no predefined workflow or iteration loop.

Leveraging the established correlation between LLM-based evaluators and human expert judgements (Lu et al., 2024), we employed a dedicated pairwise LLM evaluator, rather than direct scoring of individual hypotheses.

Each of the systems was given 100 inputs spanning a diversity of topics, from oncology to ageing and cellular senescence, generating one hypothesis per input. For every pair of generated hypotheses from different configurations, an LLM judge selected its preferred candidate (or declared a tie) for each of four metrics: *novelty*, *relevance*, *verifiability* and *significance* (from Qi et al., 2024).

We fitted a Bradley–Terry (1952) paired comparison model to the results, given by

$$\log \text{odds}(i \text{ beats } j) = \beta_i - \beta_j + \alpha z_{ij}, \tag{1}$$

where $i$ and $j$ denote 'players' or LLM configurations, $\beta_i$ is the *ability score* of the $i^{\text{th}}$ LLM system and $\alpha$ is a 'home advantage' parameter, estimating the possible order effect bias, with $z_{ij} = 1$ if the hypothesis of $i$ is presented to the evaluator first in the pair, and $z_{ij} = -1$ otherwise. Ties are treated as a half-win for each player. The continuous latent ability scores for each configuration and each metric provide a univariate ranking that captures relative performance.

The fitted ability scores and their 95% comparison intervals (calculated using a quasi-variance approximation; Firth, 2004) are visualized in Figure 4. The model shows a clear, progressive improvement in hypothesis novelty and significance as components are added to the system. The single LLM baseline was least preferred. Introducing a multi-agent structure provided significant improvement, suggesting a benefit from role-based reasoning alone. Performance further increased with the addition of external tools and with iterative refinement. Between the two, tool use provided a greater benefit, emphasizing the value of grounding the generation process in external knowledge sources. Finally, the full BioDisco system was most preferred for these metrics, demonstrating a synergistic effect between dual-mode evidence and iterative refinement, validating our approach.

Interestingly, patterns for relevance and verifiability were less clearly defined: pairs of configurations have overlapping comparison intervals, with no clear winner. While a single LLM may even be preferred for

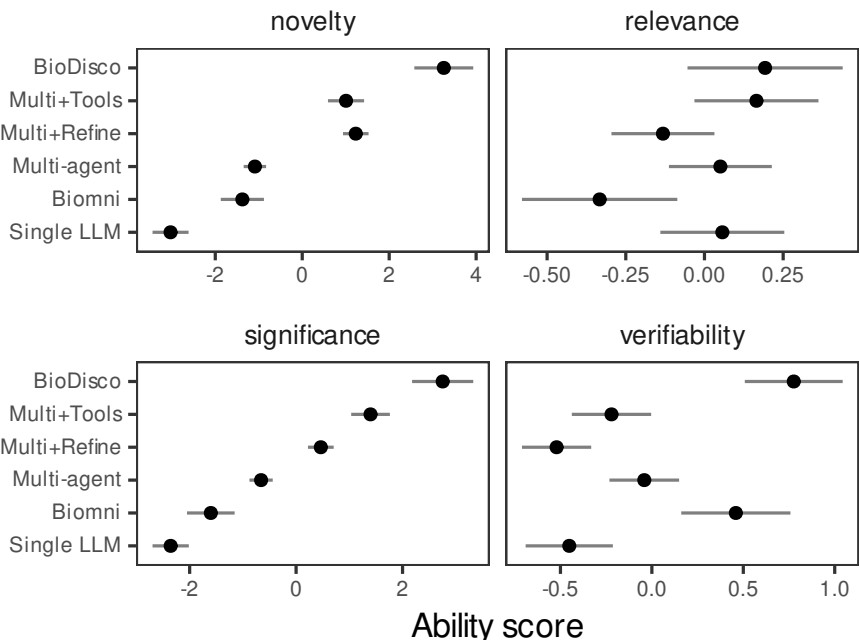

Figure 4: Centipede plot of ability scores for BioDisco, four ablation configurations and Biomni, with 95% comparison intervals. A multi-agent system clearly outperforms a single LLM (GPT-4.1) generating novel, significant hypotheses; tool use (i.e. KG and literature search) and iterative refinement each yield further improvements. Biomni, a competing system, is mostly better than a single LLM and produces verifiable hypotheses, but less novel and significant than the full BioDisco framework

Table 3: Parameter estimates $\hat{\alpha}$ from the fitted Bradley–Terry models for each of the four evaluation metrics

| Metric | Estimate | Std. Error | Statistic | $p$-value |
|---|---|---|---|---|
| Novelty | −0.10 | 0.21 | −0.51 | 0.61 |
| Relevance | 0.26 | 0.11 | 2.38 | 0.02 |
| Significance | 0.09 | 0.17 | 0.52 | 0.60 |
| Verifiability | 1.11 | 0.13 | 8.68 | 0.00 |

topical relevance, this may simply suggest a lack of external knowledge causes such a zero-shot system to remain close to the user input, at the expense of generating novel or significant hypotheses.

Estimates for the order effect parameter $\hat{\alpha}$ are given in Table 3 and suggest a statistically significant order bias for Verifiability evaluations (in favour of the item appearing first in the comparisons) but not enough evidence to indicate such an effect exists in Novelty, Relevance or Significance comparisons at the 5% level of significance.

Additional results—including a direct scoring evaluation—are provided in the Appendix.

### 3.3 Human expert evaluation

To complement the model-based assessments, and to ascertain the practical utility of BioDisco's outputs, our final evaluation involved two groups of human domain experts.

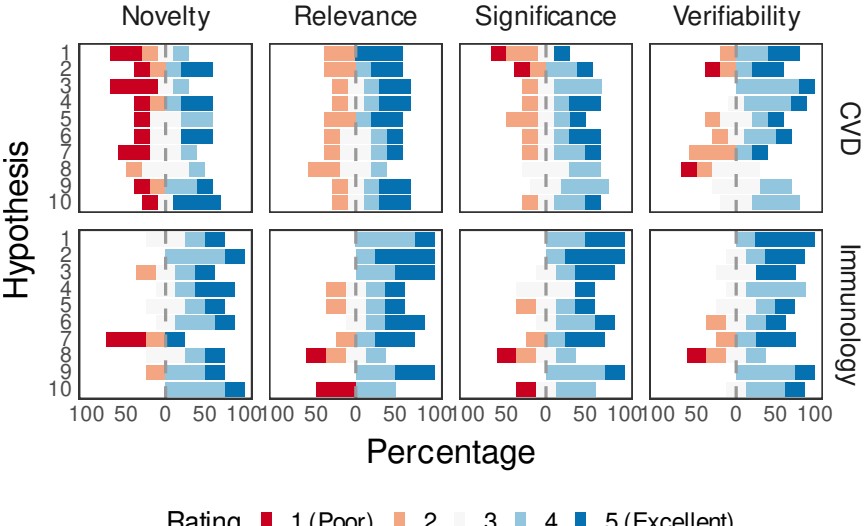

Figure 5: Ratings given by two independent groups of human experts to 10 hypotheses generated for respective topics of cardiovascular disease (CVD) and immunology

### 3.3.1 Methodology

We recruited nine experts from two biomedical fields—four in immunology and five in cardiovascular medicine—to complete an online survey about a series of topical hypotheses generated by BIODISCO. The hypothesis pairs were generated as follows.

For immunology, BIODISCO was provided with the complex prompt: "T Cell Exhaustion Mechanisms and Therapeutic Targets in NSCLC". The system was run five times to generate five distinct hypotheses, to explore its capacity to explore a single topic in depth. For cardiovascular disease (CVD), BIODISCO was provided five unique prompts related to CVD, to assess performance across different topics. One initial and one refined hypothesis was generated for each input.

For every sub-topic, experts were presented with both the initial version (the first complete hypothesis generated) and the final version (after iterative refinement). Experts independently rated each hypothesis on a 1–5 scale according to the four metrics: novelty, relevance, significance and verifiability. They also rated their confidence in their assessments and provided qualitative feedback.

### 3.3.2 Analysis

Survey responses were modelled using a single polytomous Rasch model, implemented as a Bayesian cumulative probit mixed effects model (Bürkner, 2021) for all four metrics—one for each group of experts:

$$P(Y_{ijm} \leq k) = \Phi(\tau_k - \eta_{ijm}), \tag{2}$$

where $Y_{ijm}$ denotes the ordinal rating ($k = 1, \ldots, 5$) given by rater $i$ to hypothesis $j$ on metric $m$, $\Phi$ denotes the standard normal distribution function and $\tau_k$ are the latent thresholds that partition the ordinal rating scale. The linear predictor $\eta_{ijm}$ contains both fixed and random effects:

$$\eta_{ijm} = \underbrace{\beta_m}_{\text{metric effect}} + \underbrace{u_i}_{\text{rater effect}} + \underbrace{v_{jm}}_{\text{hypothesis-on-metric effect}}, \tag{3}$$

where $\beta_m$ is a fixed effect representing the average quality or 'easiness' for metric $m$, $u_i \sim \mathcal{N}(0, \sigma_u^2)$ is a rater-specific random effect capturing overall leniency or severity of rater $i$, and $v_{jm} \sim \mathcal{N}(0, \sigma_v^2)$ is a hypothesis- and metric-specific random effect reflecting the relative quality of hypothesis $j$ on metric $m$. By disentangling rater- and metric-specific biases and accounting for the ordinal nature of responses, the model provides a

more robust and unbiased estimate of the underlying quality of each hypothesis, while pooling information across the different questions.

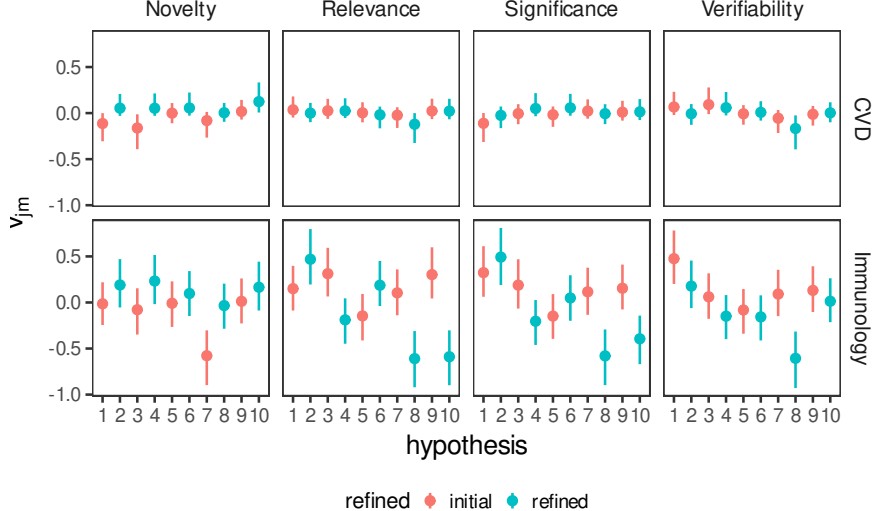

Figure 6: Posterior distributions of hypothesis-on-metric effect, $\hat{v}_{jm}$, from human evaluations of hypotheses generated by BIODISCO for cardiovascular disease and immunology

### 3.3.3  Results

Figure 5 presents the raw distribution of expert ratings. We observe that for the immunology domain, experts consistently provided high scores for BIODISCO-generated hypotheses. In contrast, in CVD, experts had more varied opinions on the novelty of some hypotheses.

The posterior distributions of the hypothesis-on-metric effects are shown in Figure 6. A clear improvement in novelty, as assessed by expert raters, is evident following the iterative refinement process. Interestingly, however, the refined hypotheses are not necessarily judged to be more verifiable—an observation that aligns our ablation study.

Qualitative feedback reinforced the system's efficacy in generating scientifically valuable and contextually relevant hypotheses. While a minor fraction of the generated hypotheses exhibited limited novelty or proposed intricate, unvalidated connections, the overarching expert consensus underscored both the scientific plausibility and practical experimental tractability. For example, one expert commented:

> "This idea could be tested with targeted experiments like activating or inhibiting GPR153 in VSMCs, running transcriptomic and ChIP-seq analyses to track CEBPB/NRF1/CD7 activity, and using in vivo vascular injury models with genetic knockouts. If these links hold up, the network could point to new therapeutic targets for preventing neointima formation."

## 4  Python Package

The BIODISCO framework is available to biomedical researchers in the form of a Python package, which can be installed via PyPI.org using

```
pip install biodisco
```

or from the code provided in the supplement. Users can customize their choice of LLM and knowledge graph through standard interfaces. Documentation is provided online.

# 5 Discussion & Conclusion

We presented BioDisco, a novel multi-agent framework for grounded and refined biomedical hypothesis discovery, made openly available for research use. A temporal evaluation on held-out data shows the ability of the system to extrapolate beyond its training corpus. An ablation demonstrates integration of iterative refinement and external knowledge improve the system more than the sum of their parts for generating 'novel', 'significant' hypotheses, and the system has been validated by human experts while using probabilistic psychometric models to account for bias and uncertainty.

## 5.1 Limitations

Hypotheses were rated as significant and novel but apparently at the expense of relevance and verifiability. One explanation for this could be an inherent trade-off between novelty and verifiability: things that are harder to test are less likely to have already been tested. Additionally, a single LLM judge may not be the best evaluator of experimental feasibility; this is explored further by a dedicated hypothesis *testing* framework, e.g. Popper (Huang et al., 2025a). Experimental validation in the laboratory is, unfortunately, beyond the scope of this paper.

Similarly, weakening 'relevance' to an input topic may indicate breadth of exploration and diversity of knowledge, or simply a limitation of the metrics proposed by Qi et al. (2024).

Our metrics-based evaluation is limited to the judgement of humans and LLMs of whether hypotheses *seem* novel or significant. A focus on self-evaluation and refinement may invoke Goodhart's Law, amplifying model artifacts and exaggerating the system's practical relevance.

A core problem of any temporal evaluation is that it assumes discoverability is equivalent to predictability, but a hypothesis can be a valid, valuable contribution to a field even if based on a flawed premise or is ultimately disproven. This makes it a poor proxy for a system's quality, as a good system should generate novel, plausible hypotheses—not just those that align with future discoveries.

In our TruthHypo evaluation, we excluded the 'no relation' class to focus on the system's ability to distinguish positive from negative associations. However, in a real-world setting, the ability to abstain may prevent hallucinations. Currently, BioDisco incorporates no such low-confidence filter.

## 5.2 Future work

While our evaluation demonstrates BioDisco's superior novelty and significance to generalist agents like Biomni (see Figure 4), the comparison does not necessarily disentangle effects of architectural design from more mundane aspects, such as data and knowledge graph availability and the relative strength of different LLMs used in agents or their evaluation. This, like any type of evaluation of LLMs trained on public data, can also be vulnerable to data leakage (see, e.g. Zhou et al., 2023). Furthermore, a purely *in silico* comparison may only test semantic plausibility; a real-world wet lab evaluation would definitively validate the utility of the system.

A major open challenge for biomedical discovery lies in the integration of diverse multi-modal biomedical data, such as clinical records, omics and medical imaging. Furthermore, more research is needed into interpretability of agentic systems, leaving traces to explain sources of evidence-based reasoning and better quantify epistemic uncertainty.

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

# Supplementary materials

This document forms the appendices for the manuscript "BIODISCO: *Multi-agent hypothesis generation with dual-mode evidence, iterative feedback and temporal evaluation.*"

## A  A Detailed Case Study

To provide a robust and unbiased assessment of BIODISCO's generated hypotheses, we conducted a structured case study combining automated outputs with expert review. The case is grounded in a recent study linking G-protein coupled receptors (GPCRs) to increased cardiovascular risk (Kobilka & Lefkowitz, 2024), and centres on two entities: the gene **GPR153** and the disease **vascular injury**. We provided "Role of GPR153 in vascular injury and disease" as input to BIODISCO. Relevant entities were drawn from the input query and used to access the literature interface to retrieve relevant publications.

Initially, the BACKGROUND agent synthesised a summary highlighting the dual role of GPR153 in vascular injury and inflammation from the retrieved literature. The EXPLORER analyzed the background summary to retrieve a subgraph consisting of genes like GPR153, CEBPB, GRN, CDK4, TTR, YAP1, SCAMP1 as well as drugs like Acamprosate and conditions such as camptodactyly.

The SCIENTIST received both the background summary and the KG subgraph based on which it proposed an initial hypothesis:

> "GPR153 activation in vascular smooth muscle cells enhances pro-inflammatory gene expression via the YAP/TAZ pathway, promoting neointima formation following vascular injury"

Subsequently, the initial hypothesis and relevant keywords were used to query the literature interface for relevant PubMed publications to find evidence for the generated hypothesis. This search identified studies such as Shao et al. (2025) which conveys how GPR153 is an orphan receptor that facilitates expression of pro-inflammation and pro-proliferation genes in smooth muscle cells by regulating cAMP levels in cells and thereby contributing to inflammation and vascular remodelling.

The initial hypothesis by SCIENTIST was evaluated by the CRITIC, considering the background and literature. Novelty was rated 4, reflecting the novel aspect of linking GPR153 activation to the YAP/TAZ pathway and neointima formation in vascular smooth muscle cells. Relevance was given a 5 due to the significant interest in vascular injury in therapeutic research. Significance was rated 4, acknowledging the therapeutic potential of targeting this pathway. This gave the initial hypothesis 17 out of 20.

To improve the hypothesis, the REVIEWER first focused on Novelty by highlighting "lack of mechanistic insight" on how exactly GPR153 activation influences the YAP/TAZ pathway. It decided to obtain additional knowledge graph evidence, and therefore used the current hypothesis, critic feedback, and background information to query the KG interface. This yielded a new subgraph from the knowledge graph with GPR153, YAP, and TAZ as the key entities. Using this new information, the REFINER then reformulated the hypothesis to:

> "GPR153 activation in vascular smooth muscle cells enhances pro-inflammatory gene expression by promoting CEBPB-mediated YAP1 signalling, thereby potentially integrating with EGR1 and GSK3B pathways to exacerbate neointima formation following vascular injury"

Following this, the system re-entered the feedback loop. Incorporating newly identified literature, the CRITIC raised the Novelty and Significance scores to 5, while Relevance remained at 5 and Verifiability at 4, yielding an overall score of 19. Normally, iteration would stop here, as further improvement is constrained by the inherent complexity of the biology rather than lack of evidence. For demonstration, however, we continued the process to illustrate how the system explores more intricate mechanistic hypotheses. Additional literature

searches for each key gene and their combinations further substantiated the potential connections identified in the evolving hypothesis. Subsequently, the REVIEWER focused on Verifiability—the only criterion not awarded a top score. Incorporating new subgraph information and additional literature, the REFINER further elaborated the hypothesis to include a broader CEBPB-mediated network involving YAP1, EGR1, and GSK3B:

> "GPR153 activation in vascular smooth muscle cells enhances pro-inflammatory gene expression by facilitating CEBPB-mediated network involving YAP1, EGR1, and GSK3B, creating a complex signalling cascade that drives neointima formation after vascular injury"

In the next round, the CRITIC again assigned top scores for Novelty, Significance, and Relevance, while Verifiability remained at 4, reflecting the growing experimental challenges posed by increased mechanistic complexity. To further demonstrate system capabilities, additional genes (NRF1, CD7, and GSK3B) were integrated, supported by targeted literature searches:

> "GPR153 activation in vascular smooth muscle cells enhances pro-inflammatory gene expression through a CEBPB-mediated network, integrating NRF1 and CD7 interactions with YAP1 and GSK3B, thereby orchestrating a multifaceted signalling cascade that drives neointima formation following vascular injury"

For this, CRITIC gave Novelty a 5 because it "combines several key regulatory proteins" to provide a comprehensive signalling network. Both Relevance and Significance were also scored at 5. Verifiability was still at 4 because CRITIC noted that although techniques like gene editing, pathway analysis and in vivo models of vascular injury could be used to test the hypothesis, the "complexity of the interactions may pose challenges in definitively confirming all proposed relationships". This resulted in an overall score of 19/20.

## B    Additional Results

### Examples from Qi et al. (2024)

In Table 4, we present selected examples comparing gold-standard hypotheses with those generated by BIO-DISCO, along with their cosine similarity scores (see main paper for details on temporal evaluation). While Figure 3 demonstrates that BIODISCO produces hypotheses with higher semantic similarity than unrelated gold pairs, this table illustrates whether that similarity arises from shared terminology or meaningful mechanistic insight.

In the first example, we see that BIODISCO accurately identifies the core mechanistic insight present in the gold hypothesis that that inhibiting VEGFR2 in hypertrophic chondrocytes interrupts ERK1/2 activation and subsequent apoptosis. We can also see how BIODISCO provides more nuanced insights by integrating additional molecular pathways when compared to the gold-standard hypothesis.

Similarly, in the second example, BIODISCO successfully proposes and extends the insight from the gold-standard hypothesis, adding specific details about potential pathways and proteins involved in the process. Unlike the previous examples where the generated and gold-standard hypotheses shared the same mechanistic insights, this third pair takes different directions, resulting in a low similarity score.

### TruthHypo

Table 5 presents the results for each class across the three tasks in the TruthHypo dataset. The high Precision, Recall, and $F_1$ scores achieved by BIODISCO demonstrate its ability to accurately identify the relationships between the given entities.

### Direct Evaluation using LLMs

Table 6 reports LLM-based evaluation scores across four metrics - novelty, relevance, significance, verifiability, and an overall score. As expected, we observe consistent improvements when moving from a single-agent

Table 4: Examples of gold hypotheses and hypotheses generated by BioDisco, with cosine similarity, $S_{ij}$, computed from their respective embeddings.

| Gold hypothesis | BioDisco | $S_{ij}$ |
|---|---|---|
| Inhibition of VEGFR2 should interrupt phosphate-induced ERK1/2 activation and subsequent apoptotic events in hypertrophic chondrocytes. | Inhibition of VEGFR2 in hypertrophic chondrocytes suppresses ERK1/2 activation, preventing apoptosis by modulating downstream pathways involving PPP2R2A or CORO1C, thereby ameliorating hypophosphatemic rickets. | 0.86 |
| PPFIBP1 may play a role in the development of chemoresistance in MM. | Elevated PPFIBP1 upregulation in multiple myeloma cells enhances bortezomib resistance by activating NF-$\kappa$B signaling, supported by PPFIBP1's role in promoting RelA stability and cyto-nuclear translocation, indicating a direct link to chemoresistance. | 0.65 |
| TkR86C expression in damaged wing discs and brain suggests a possible non-cell-autonomous role in regeneration | Neuregulin signaling via the AMPK/mTOR pathway enhances progenitor cell proliferation in limb regeneration. | 0.39 |

Table 5: Performance of the proposed system by group, reporting Precision, Recall, F1, Accuracy, and Support for each relation and macro average.

| Group | Precision | Recall | $F_1$ | Acc | Support |
|---|---|---|---|---|---|
| Chemical-Gene / negative | 0.917 | 0.880 | 0.898 | | 50 |
| Chemical-Gene / positive | 0.885 | 0.920 | 0.902 | | 50 |
| Chemical-Gene (avg) | 0.901 | 0.900 | 0.900 | 0.900 | 100 |
| Disease-Gene / inhibit | 0.848 | 0.780 | 0.812 | | 50 |
| Disease-Gene / stimulate | 0.796 | 0.860 | 0.827 | | 50 |
| Disease-Gene (avg) | 0.822 | 0.820 | 0.820 | 0.820 | 100 |
| Gene-Gene / negative | 0.867 | 0.780 | 0.821 | | 50 |
| Gene-Gene / positive | 0.800 | 0.880 | 0.838 | | 50 |
| Gene-Gene (avg) | 0.833 | 0.830 | 0.830 | 0.830 | 100 |
| ALL DATA (avg) | 0.844 | 0.842 | 0.842 | 0.850 | 300 |

baseline to a multi-agent setup and further to BioDisco. However, the performance margins between versions remain narrow. Also, the effect of access to external interfaces is particularly unclear, with only minor differences in scores.

Table 6: Direct comparison with a evaluator LLM and various ablation configurations of the BIODISCO framework: with and without tools (i.e. KG and literature search), iterative refinement and multi-agentic reasoning. The baseline is a single-agent LLM only. Reported mean ± standard deviation of scores for a set of 100 generated hypothesis.

| Configuration | Novelty | Relevance | Significance | Verifiability | Overall |
|---|---|---|---|---|---|
| GPT-4.1 (baseline) | 1.38 ± 0.60 | 3.00 ± 0.00 | 1.89 ± 0.72 | 2.82 ± 0.22 | 2.27 ± 0.31 |
| Multi-agent system | 2.06 ± 0.32 | 3.00 ± 0.00 | 2.48 ± 0.10 | 2.70 ± 0.27 | 2.56 ± 0.09 |
| Multi-agent + tools | 2.43 ± 0.18 | 3.00 ± 0.00 | 2.50 ± 0.00 | 2.46 ± 0.22 | 2.60 ± 0.06 |
| Multi-agent + refine | 2.46 ± 0.14 | 2.99 ± 0.05 | 2.50 ± 0.00 | 2.44 ± 0.24 | 2.60 ± 0.07 |
| BIODISCO | 2.54 ± 0.14 | 2.99 ± 0.05 | 2.55 ± 0.14 | 2.54 ± 0.42 | 2.66 ± 0.13 |

These observations highlight that scalar ratings alone may be insufficient to capture more nuanced improvements in hypothesis quality. This underscores the value of pairwise evaluation, where an LLM compares hypotheses directly and can better distinguish subtle but meaningful differences in structure, specificity, and insight that may be overlooked in absolute scoring schemes.

## C   Paired Comparison Models

Paired comparison models were fitted using the R package `BradleyTerry2` (Turner & Firth, 2012), with quasi-variance approximations computed using `qvcalc` (Firth, 2025). The Bradley–Terry model (Bradley & Terry, 1952) is given by

$$\log \text{odds}(i \text{ beats } j) = \beta_i - \beta_j + \alpha z_{ij}, \tag{4}$$

where $i$ and $j$ denote 'players' or LLM configurations, $\beta_i$ is the *ability score* of the $i^{\text{th}}$ LLM system and $\alpha$ is a 'home advantage' parameter, estimating the possible order effect bias, with $z_{ij} = 1$ if the hypothesis of $i$ is presented to the evaluator first in the pair, and $z_{ij} = -1$ otherwise.

For our model, ties were awarded as half wins to each player. An extension of the Bradley–Terry model that explicitly supports ties, the Davidson (1970) model, was also fitted separately, but yielded very similar results (not presented here).

Quasi-variances (Firth, 2004) aim to minimize the squared loss

$$\min \sum_{i<j} (q_i + q_k - v_{ij})^2, \tag{5}$$

where $q_i$ is the quasi-variance of player $i$ and $v_{ij}$ is the covariance term (i.e. off-diagonal entries between players $i$ and $j$. The package `qvcalc` returns relative errors as a measure of the quality of this approximation: the distribution of relative errors is given in Figure 7 as a beeswarm plot (Selby, 2020) and appear to be mostly acceptable.

## D   Compute Infrastructure

All experiments were executed on a commodity CPU server (4 cores, 16 GB RAM, no GPU), making the workflow straightforward to deploy and reproduce. Inference time and cost depend chiefly on (i) the number of refinement iterations, (ii) the size of the knowledge-graph (KG) and literature evidence retrieved, (iii) PubMed/Neo4j access latency, and (iv) network latency to the OpenAI GPT-4.1 API.

For all LLM-based agents, random seed (cache seed) was fixed to 42 to support reproducibility, and temperature was set between 0.2 and 0.5 depending on the agent role to balance generation diversity and stability.

Four configurations were benchmarked in Table 7. The single-pass multi-agent baseline without KG or literature ("Multi-agent") is the most economical, averaging $0.004 per hypothesis. In contrast, the full BioDisco pipeline, which involves three refinement iterations, had the highest cost of approximately 0.071 US dollars. Incorporating KG and literature significantly increases token usage and API cost but provides

richer contextual and mechanistic evidence. On average, BioDisco completes a full inference in 2 to 3 minutes per hypothesis, while the lightweight baseline finishes within one minute.

Table 7: Computation cost under different system settings. All results are based on experiments using GPT-4.1. API cost in United States dollars ($) is reported per hypothesis.

| Setting | Input Tokens | Output Tokens | US $ |
|---|---|---|---|
| Multi-agent | 1393 | 264 | 0.004 |
| Multi+Tools | 8113 | 653 | 0.017 |
| Multi+Refine | 30263 | 1495 | 0.019 |
| BioDisco | 72828 | 4424 | 0.071 |

All essential components (OpenAI LLM API, Neo4j KG, and open-source retrieval/embedding libraries) are publicly available.

## E    Human evaluation

### Survey Design

The human evaluation of hypotheses took place online via a bespoke Microsoft Form. Dissemination of the survey was through direct and e-mail communication and through a consortium-wide mailing list.

Participants had the right to remain anonymous or to be acknowledged by name in subsequent publications. They were asked to give their institutional affiliation, years of experience, and discretized number of publications in their field of expertise. For each hypothesis,

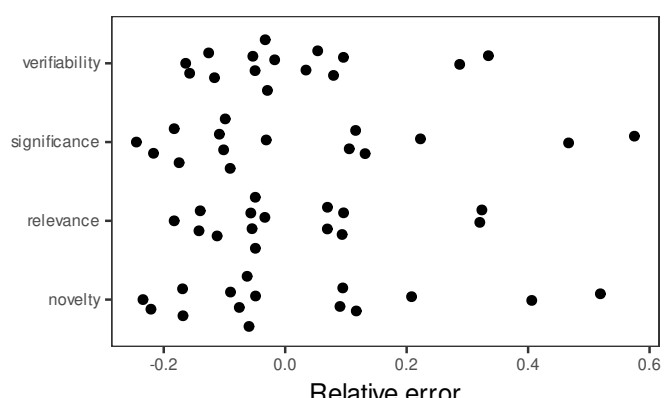

Figure 7: Beeswarm plot showing distribution of relative errors from the quasi-variance approximation

the participant was asked to rate it for novelty, significance, verifiability and relevance (to the given input context), each on a scale from 1–5, where 1 is the lowest and 5 is the highest score. The set of metrics is based on Qi et al. (2024) and the descriptions were the same as given to LLM evaluators (see Prompt 8).

Additionally, each respondents could score their level of confidence in their chosen ratings for each hypothesis (on a 1–5 scale) according to their familiarity with the topic area, as well as add free-text comments. All fields in the survey were optional, so the user could skip giving a rating for any part of any hypothesis, for any reason. This was to reduce the risk of unconfident or uninterested respondents giving arbitrary scores. A sample from the online questionnaire is presented in Figure 8.

### Evaluation Material

We provide the full list of topics, initial hypotheses, and final hypotheses used for human evaluation in both cardiovascular disease (CVD) and immunology domains (see Table 8 and Table 9).

### Bayesian Item Response Modelling

A summary of this model appears in § 3.3 ("What do human domain experts think?") of the main text, and we provide the full details here. To quantitatively assess human evaluation scores while accounting for both rater and hypothesis-specific variability, we fit a Bayesian cumulative ordinal mixed-effects model to

## Topic: Role of GPR153 in vascular injury and disease

**Please give a score to the below hypotheses for the following metrics**

- **Novelty:** Assesses whether the proposed mechanism, strategy, or relationship introduces an idea not found current mainstream literature. High novelty means the hypothesis suggests a new entity, pathway, or relationship previously unreported. Mere rewording or minor variants of known mechanisms should score low.
- **Relevance:** Evaluates the degree to which the hypothesis logically fits and aligns with the facts, themes, and context given in the **topic**. Hypotheses off-topic, not directly grounded in the context, or focusing on unrelated mechanisms should score low.
- **Significance:** Reflects the potential impact of the hypothesis if true, both scientifically (advancing fundamental understanding) and/or clinically (improving diagnosis, treatment, or patient outcomes). Trivial extensions or already well-established ideas score low; potential paradigm shifts or major therapeutic implications score high.
- **Verifiability:** Judges whether the hypothesis can be feasibly tested using currently available experimental or computational methods, in accordance with ethical standards. Hypotheses requiring infeasible technology or unethical human/animal experimentation should score low; those testable with standard assays/models score high.

7. Hypothesis 1: GPR153 activation in vascular smooth muscle cells enhances pro-inflammatory gene expression via the YAP/TAZ pathway, promoting neointima formation following vascular injury.

Note: Rate each metric on a 1–5 scale, where **1 = Poor** and **5 = Excellent**.

|  | 1 | 2 | 3 | 4 | 5 |
|---|---|---|---|---|---|
| Novelty | ○ | ○ | ○ | ○ | ○ |
| Relevance | ○ | ○ | ○ | ○ | ○ |
| Significance | ○ | ○ | ○ | ○ | ○ |
| Verifiability | ○ | ○ | ○ | ○ | ○ |

8. Comments

> Enter your answer

9. How confident are you about your scoring?

|  | 1. No relevant background; highly uncertain. | 2. Different field or limited familiarity; low confidence. | 3. In the field but not highly experienced; moderate confidence. | 4. In the field with strong understanding; high confidence. | 5. Expert or highly knowledgeable; extremely confident. |
|---|---|---|---|---|---|
| Confidence | ○ | ○ | ○ | ○ | ○ |

Figure 8: Screenshot from the expert evaluation questionnaire, showing a single hypothesis and scoring rubric

Table 8: CVD–related human evaluation cases, showing the input, initial hypothesis, and final (refined) hypothesis produced by the system.

| Input | Initial Hypothesis | Final Hypothesis |
|---|---|---|
| Role of GPR153 in vascular injury and disease | GPR153 activation in vascular smooth muscle cells enhances pro-inflammatory gene expression via the YAP/TAZ pathway, promoting neointima formation following vascular injury. | GPR153 activation in vascular smooth muscle cells enhances pro-inflammatory gene expression through a CEBPB-mediated network, integrating NRF1 and CD7 interactions with YAP1 and GSK3B, thereby orchestrating a multifaceted signaling cascade that drives neointima formation following vascular injury. |
| FDX1 and cholesterol metabolism in cardiovascular risk | Elevated expression of FDX1 in macrophages enhances cholesterol efflux, thereby reducing carotid intima media thickness and lowering cardiovascular risk through modulation of inflammatory pathways. | Elevated FDX1 expression in macrophages enhances cholesterol efflux and reduces carotid intima media thickness by engaging regulatory networks involving EGR1, NR4A2, and CAPZA2, with experimental validation through gene expression analyses and metabolic profiling to disentangle these interactions and their impact on cardiovascular risk. |
| PTLOs and immune responses in atherosclerosis | Activation of PTGDS within tertiary lymphoid organ-like structures enhances B cell-mediated antibody production, promoting plaque instability in atherosclerosis. | Activation of PTGDS within tertiary lymphoid organ-like structures not only enhances B cell-mediated antibody production but also establishes a feedback loop involving TNF-$\alpha$ and AKT1 that regulates macrophage efferocytosis, thereby orchestrating inflammatory responses that escalate plaque instability in atherosclerosis. |
| Rare pathogenic variants in G-protein-coupled receptor genes for atherosclerosis | Rare pathogenic variants in GPCR genes enhance inflammatory signaling, increasing the risk of atrial fibrillation and cardiomyopathy in patients independent of traditional cardiovascular risk factors. | Rare pathogenic variants in GPCR genes disrupt inflammatory signaling and gut microbiota interactions, which synergistically exacerbate atrial fibrillation and cardiomyopathy, suggesting a novel approach for identifying targeted biomarkers that link these pathways in personalized cardiovascular disease management. |
| Cpt1b and heart regeneration | Inhibition of CPT1B enhances cardiomyocyte proliferation by reactivating cardiogenic factors and suppressing NF-$\kappa$B-mediated inflammatory responses during heart regeneration. | Inhibition of CPT1B enhances cardiomyocyte proliferation through a regulatory cascade involving GOLT1A that modulates mitochondrial dynamics and lipid metabolism, while concurrently suppressing NF-$\kappa$B signaling and promoting a metabolic shift, thereby facilitating heart regeneration. |

Table 9: Immunology–related human evaluation cases, showing the input, initial hypothesis, and final (refined) hypothesis produced by the system.

| Input | Initial Hypothesis | Final Hypothesis |
|---|---|---|
| Investigation of molecular drivers underlying T cell exhaustion in non-small cell lung cancer (NSCLC), with a focus on identifying novel, druggable targets to enhance the efficacy of immune checkpoint inhibitor (ICI) therapies. | Inhibition of STK31 in NSCLC tumor microenvironments enhances CD8 T-cell functionality and mitigates T-cell exhaustion, improving the efficacy of immune checkpoint inhibitors. | Inhibition of STK31 in NSCLC tumor microenvironments enhances CD8 T-cell functionality by disrupting ETS1 and PIK3R1-mediated immunosuppressive pathways, while promoting pro-inflammatory cytokine release, thereby synergistically improving responsiveness to immune checkpoint inhibitors and overcoming T-cell exhaustion. |
| | Inhibition of PIK3R1-CTLA4 interaction attenuates T cell exhaustion and enhances the efficacy of immune checkpoint inhibitor therapy in non-small cell lung cancer. | Adoptive transfer of NSCLC patient T cells engineered via CRISPR to disrupt the PIK3R1-CTLA4 axis and enhance CD7 costimulation will yield durable reversal of exhaustion and superior clinical responses to immune checkpoint inhibitor therapy in early-phase clinical trials. |
| | PPP2CA suppresses STAT3-mediated cytokine signaling in the NSCLC tumor microenvironment, thereby reducing CD8+ T cell exhaustion and enhancing the efficacy of immune checkpoint inhibitors. | PPP2CA dephosphorylates STAT3 and modulates MYC and ETS1 activity, leading to altered TNFSF4 signaling and a reprogrammed cytokine milieu in NSCLC that diminishes CD8+ T cell exhaustion and enhances the therapeutic efficacy of immune checkpoint blockade. |
| | CTLA4 upregulation in tumor-infiltrating T cells promotes T cell exhaustion and resistance to immune checkpoint inhibitors in non-small cell lung cancer. | In NSCLC, lactate-induced activation of SRPK1 enhances USP39-mediated RNA splicing in regulatory T cells, leading to CTLA4 upregulation and T cell exhaustion, with SRPK1 or USP39 inhibition predicted to restore antitumor immunity and sensitize tumors to immune checkpoint inhibitors. |
| | Inhibition of CK2B enhances PIK3R1-mediated CTLA4 downregulation, reducing T cell exhaustion and improving immune checkpoint inhibitor efficacy in non-small cell lung cancer. | Inhibition of CK2B and PIK3R1, in conjunction with IL-21R activation, enhances CTLA4 downregulation and STAT5B-mediated T cell reactivation, improving immune checkpoint inhibitor responses in non-small cell lung cancer by targeting progenitor-exhausted T cells within the tumor microenvironment. |

the ordinal scores assigned by each rater. Let $Y_{ijm}$ denote the ordinal rating (on a scale of 1 to 5) given by rater $i$ to hypothesis $j$ on metric $m$. The model estimates the cumulative probability for each rating category $k \in \{1, 2, 3, 4\}$ as

$$P(Y_{ijm} \leq k) = \Phi(\tau_k - \eta_{ijm}), \tag{6}$$

where $\Phi$ denotes the standard normal cumulative distribution function (probit link), and $\tau_k$ are the latent thresholds that partition the ordinal rating scale.

The linear predictor $\eta_{ijm}$ contains both fixed and random effects:

$$\eta_{ijm} = \underbrace{\beta_m}_{\text{metric effect}} + \underbrace{u_i}_{\text{rater effect}} + \underbrace{v_{jm}}_{\text{hypothesis-on-metric effect}}, \tag{7}$$

where $\beta_m$ is a fixed effect representing the average quality or 'easiness' for metric $m$, $u_i \sim \mathcal{N}(0, \sigma_u^2)$ is a rater-specific random effect capturing overall leniency or severity of rater $i$, and $v_{jm} \sim \mathcal{N}(0, \sigma_v^2)$ is a hypothesis-and metric-specific random effect reflecting the relative quality of hypothesis $j$ on metric $m$.

Using `brms` (Bürkner, 2021) we fit a single model across all metrics to increase statistical power, under the assumption that the rater leniency effects are constant across metrics. This approach pools information across all dimensions for more robust estimation of both rater and hypothesis effects.

## F   Agent Roles

Here we describe in detail the role of the individual agents, their respective prompts and the technical implementation of interactions with the knowledge graph and PubMed.

### F.1   Implementation Details

**Adaptive adjustment of KG query parameters**

First, the upstream agent (e.g. ReviewerAgent) automatically identifies suboptimal evaluation criteria based on task objectives and intermediate feedback, and employs LLM-based reasoning to generate updated traversal depth (`DEPTH_OVERRIDE`) and relation type lists (`RELS_OVERRIDE`). Second, the downstream KG interface merges default relation types and maximum traversal depth for the specified domain of the hypothesis (e.g. the molecular domain includes `protein_protein`, `molfunc_protein` and `bioprocess_protein` relations by default). If override parameters are provided, they fully replace the defaults, directly controlling both single-hop and multi-hop KG retrieval. If the result exceeds a predefined threshold, the system automatically reverts to the domain default settings and applies top-$k$ pruning. This mechanism ensures that relation types and traversal depth are adaptively optimized across domains in response to task requirements and real-time feedback.

**Standardized representation of KG subgraphs**

The KG interface returns subgraphs as structured JSON objects containing nodes, direct edges, and multi-hop paths. Each node entry includes a unique identifier, name, type, and standardized attributes, while each edge specifies the source, target, relation type, weight and path information. For downstream use, the system summarizes and converts these subgraphs into standardized plaintext: each node is represented as 'name(type)', each direct edge as 'source(type) → relation type(weight) → target(type)' and each multi-hop path as a sequential concatenation of nodes and relations. This process produces concise, human-readable text formats suitable for subsequent LLM-based analysis or other downstream tasks.

**LLM-based PubMed querying strategy**

We employ LLM-powered agents to analyze user-provided keywords, hypotheses, and associated feedback. Through carefully designed system prompts, the LLM automatically groups semantically similar keywords into synonym sets (using OR logic within each group), and determines the optimal Boolean logic (AND or

OR) for combining these groups based on context and query intent. The LLM returns a structured JSON output specifying grouped terms, Boolean logic and multiple candidate query formulations with guidance notes.

The system then parses this structured output and generates PubMed API-compatible query strings via templated rules: each group of terms is tagged for MeSH or TIAB fields, inter-group logic is set as recommended by the LLM, and query constraints such as date range and publication type are automatically added. For each generated query, the system executes PubMed searches in sequence; if insufficient results are obtained, subgroup queries are recursively constructed. All retrieved articles are deduplicated, merged and ranked by publication date. This workflow enables robust, context-adaptive PubMed retrieval by tightly integrating LLM-driven planning with programmatic query construction and post-processing.

### F.2 Planner agent

The PLANNER agent functions as the central coordinator for the hypothesis discovery pipeline. It receives keywords input by users and manages the sequential execution of all specialized agents, including background retrieval, knowledge graph exploration, hypothesis generation, evaluation, and refinement. At each stage, the PLANNER passes relevant intermediate outputs between agents, monitors progress, and handles control flow decisions such as triggering iterative refinement or terminating the process. In addition to orchestrating agent execution, the PLANNER can optionally produce a concise research plan summarizing the workflow steps taken for a given task. This design promotes modularity, simplifies pipeline management, and facilitates user intervention or expert oversight at any stage of the discovery process.

#### Prompt 1: PLANNER agent

```
Develop a clear, stepwise research workflow based on the provided background text.
Your plan should outline :
1. Domain selection;
2. Knowledge graph retrieval steps;
3. Hypothesis generation;
4. Iterative refinement using literature and graph evidence;
5. Final decision —making.
Respond with numbered steps.
```

### F.3 Background agent

The BACKGROUND agent is responsible for constructing a concise and informative textual background for each hypothesis discovery task. It receives keywords or research topics and then retrieves relevant biomedical literatures through the literature interface, typically by querying PubMed. The agent then synthesizes and summarizes the retrieved content, producing a structured background paragraph that integrates the most pertinent findings and context. This background serves as a foundation for downstream agents, ensuring that subsequent hypothesis generation and evaluation are grounded in up-to-date and domain-relevant evidence.

#### Prompt 2: BACKGROUND agent

```
You are given a list of PubMed article metadata blocks about a specific disease and a set of core genes
    or biological entities . Write a concise , well—structured background paragraph ( less than 150 words)
    that summarizes key mechanistic insights and highlights the relationships between the core genes
    and disease—relevant biological processes (such as EMT, inflammation, senescence, signaling , etc .) .
Requirements:
1. Clearly explain how the core genes are linked to disease mechanisms, pathways, or phenotypes based on
    the literature .
2. Emphasize causal or regulatory connections when possible , rather than just listing associations .
3. Do not copy sentences verbatim from abstracts . Always synthesize and paraphrase information in your
    own words.
4. Use clear , logical , and scientifically precise language.
5. Avoid including superfluous or generic information ; focus on mechanistic insights most relevant to the
    disease and core genes.
```

### F.4  Explorer agent

The EXPLORER agent retrieves and summarizes subgraphs from a biomedical knowledge graph based on provided background information and keywords. The information passed to EXPLORER includes keywords and a text as background reference. It first maps the input keywords to candidate standardized entities in the graph, then leverages LLM to select the most contextually relevant nodes using the background as guidance. Then performs composite queries to extract related nodes and multi-hop paths based on these anchor entities. Query parameters, including hop depth, relation types, and result size are dynamically adjusted depending on the reasoning stage: broad subgraphs are retrieved during initial hypothesis generation to encourage diverse exploration, while later refinement stages focus on deeper, localized evidence addressing specific weaknesses such as low novelty or verifiability. The agent returns a structured subgraph summary that provides precise contextual support for downstream hypothesis generation and evaluation.

#### Prompt 3: EXPLORER agent

Given a background text and a list of candidate KG node, Based on the background information, select the
    most relevant nodes (5—10) to use for subgraph construction.
1. Only choose from the provided candidates.
2. Output only a JSON array of the selected.
3. Do not output any extra text, explanations, or formatting.

### F.5  Scientist agent

The SCIENTIST agent generates initial biomedical hypotheses by reasoning over the structured KG subgraph and textual background. It is fed with background information generated by BACKGROUND agent and subgraph information generated by EXPLORER agent, and then simulates the inference process of a researcher, identifying potentially novel and testable associations between entities based on mechanistic context and graph structure. Each hypothesis is expressed in natural language and describes a potential causal or regulatory relationship between biomedical entities. To encourage exploration of the hypothesis space, the ScientistAgent typically generates three candidate hypotheses in parallel, which serve as inputs for subsequent evaluation and refinement stages.

#### Prompt 4: SCIENTIST agent

Generate up to 3 concise, testable biomedical hypotheses. Each hypothesis must be grounded in both the
    background and KG context, but extend current knowledge with a novel mechanistic, causal,
    regulatory, or predictive insight.
 Guidelines:
1. Integrate both background and KG context.
2. Propose new biological mechanisms or interactions, not summaries or rephrasings of input.
3. Use precise scientific language, including mechanistic verbs such as: activates, inhibits, modulates,
    represses, etc.
4. Each hypothesis must be a single, plausible, testable sentence ($<=$ 30 words) with clear entities and
    measurable outcomes.
5. Output only the hypotheses, no numbering, bullets, explanations, citations, or evidence fields.
 Examples (good):
 Activation of TGF—$\beta$ in smooth muscle cells promotes vascular remodeling in hypertension.
 Loss of gene X enhances inflammatory response to toxin Y in liver tissue.
 Examples (bad, avoid):
 CVD is associated with Wnt signaling and fibrosis.
 Output:
 Each line should be a standalone hypothesis. Return exactly one hypothesis per line, and nothing else.

### F.6  Critic agent

The CRITIC agent provides structured evaluation of each candidate hypothesis based on supporting evidence, offering clear feedback signals to the system. Three pieces of information are passed to it simultaneously, including the background generated by BACKGROUND agent, the current hypothesis, and the corresponding relevant references (used in hypothesis generation and improvement). It scores hypotheses along four core dimensions: novelty, relevance, significance, and verifiability. Each dimension is rated on a 0–5 scale and

accompanied by a brief explanation that justifies the score. The assessment is grounded in the LLM integrated understanding of the hypothesis, background, and literature evidence. The resulting evaluations guide downstream diagnosis and refinement by identifying weaknesses and informing targeted revision strategies.

### Prompt 5: CRITIC agent

```
Assess the hypothesis using four metrics:
Novelty: Does it introduce ideas not present in the background?
Relevance: How well does it align with the background and supporting evidence?
 Significance : What is its  potential  to advance  biological  understanding  or  clinical   practice ?
  Verifiability  : Can it be  reliably  tested  with current   scientific  methods?
Rate each metric on a 0—5 scale:
0 = no merit, 1 = very  slight , 2 = slight , 3 = moderate, 4 = strong, 5 = exceptional .
Be conservative : award a 5 only  if  the hypothesis  fully  meets the  criterion  with no reservations . Provide
       one sentence of  rationale  per metric .
Output:
<Metric>: Score <X>
<One—sentence rationale>
(repeat for  all  4 metrics)
At the end, write on a separate  line :
 Overall  Score:  <value>/20
```

## F.7   Reviewer agent

The REVIEWER agent identifies weak dimensions in each hypothesis based on the scores and explanations provided by the CriticAgent. To make a sound decision, it receives full feedback from the CRITIC agent, along with the current hypothesis. It then prioritizes low-scoring criteria and, depending on the evaluation content, selectively triggers access to external knowledge sources, including the knowledge graph, literature, or background. The Agent does not directly modify the hypothesis, instead, it outputs retrieval actions along with the relevant supporting information, which are then passed to the Refiner for hypothesis revision.

### Prompt 6: REVIEWER agent

```
Given the CriticAgent 's markdown critique ( scores  0—5 with  rationales ),  the  current  hypothesis , and
       background text, recommend follow—up actions and query adjustments.
Steps:
1.  Identify  all  metrics  scoring  <= 3. If none,  select  a 4—point metric using this   priority : Novelty >
        Significance  > Relevance >  Verifiability .
2.  Recommend all relevant actions  from ['neo4j ',  'pubmed', 'background']:
   — For low novelty or  mechanistic  gaps: use 'neo4j'; add 'pubmed' if  literature  may help.
   — For low   verifiability  : use 'pubmed'; add 'neo4j ' if KG includes measurable pathways.
   — For low relevance : use 'background'.
   — If  multiple  metrics  are low, recommend all relevant  actions
3. Output exactly 3  lines :
    ACTIONS:action1,action2
    DEPTH_OVERRIDE:<integer>
    RELS_OVERRIDE:rel1,rel2,...
Rules :
Always recommend at least one action .
Include  all  actions  relevant  to low—scoring metrics .
Output must strictly  follow  the format above with no extra  explanation .
Example:
If  Novelty=2 and  Verifiability =3
ACTIONS:neo4j,pubmed
```

## F.8   Refiner agent

The REFINER agent revises and improves a given hypothesis based on received feedback and supplemental information. Specifically, it receives the low-scoring content and the integrated complementary knowledge produced by the REVIEWER agent, and the current hypothesis. It integrates low-scoring metrics and their explanations from the CRITIC Agent, along with new knowledge retrieved by the REVIEWER Agent. Using LLM synthesizes and restructures multi-source inputs to generate a revised hypothesis with enhanced

novelty, verifiability, and scientific relevance. The refinement process explicitly targets previously identified weaknesses, and the resulting hypothesis is returned to the evaluation loop for further evaluation.

Prompt 7: REFINER agent

Improve the current hypothesis based on the provided critic feedback, and new information (from Neo4j, PubMed, or background). Only make content—level changes that directly address the weaknesses.

Rules:
1. For each identified weakness, briefly state what is missing or imprecise in the hypothesis (one sentence per metric).
2. Review all new information:
    If high—quality, relevant content addresses a weakness, explain how it helps and revise the hypothesis accordingly.
    If no new information directly addresses the weaknesses, use relevant new information and scientific reasoning and the provided background to make a real, meaningful improvement, but only if this improvement is justified by the context.
    If nothing useful is found, or only stylistic edits are possible, clearly state this and leave the hypothesis unchanged.
3. Do not rephrase or reword unless it results in a real improvement. Do not invent content unsupported by evidence.

Example 1 (with helpful new info):
Step 1: The hypothesis lacks a mechanism linking Wnt inhibition to reduced fibrosis.
Step 2: New PubMed evidence suggests TGF—$\beta$ mediates this process.
Step 3: Adding TGF—$\beta$ clarifies the pathway.
Inhibition of Wnt signaling reduces cardiac fibrosis via downregulation of TGF—$\beta$ activity.
Example 2 (no helpful info):
Step 1: The hypothesis lacks a mechanistic link.
Step 2: No new information improves this.
Step 3: No justified revision possible.
Overexpression of SOD2 reduces neurodegeneration by mitigating oxidative stress in dopaminergic neurons.
Output:
    1—4 short reasoning steps (one per line)
    Final refined hypothesis as the last line (no numbering, no extra text)
Instructions:
    Use only provided context (background + new info).
    Each reasoning step must be a complete, self—contained sentence.
    Do not include explanations, citations, or bullet points.

## G    LLM Evaluators

Two LLM evaluation paradigms were used: a *direct* evaluator scores hypotheses on a numerical scale 1–5 for novelty, relevance, significance and verifiability, similar to the CRITIC agent.

Prompt 8: Direct evaluator

You are a senior biomedical reviewer.

Task:
Evaluate the following hypothesis by assigning a score for each metric (Novelty, Relevance, Significance, Verifiability) and providing a concise reason.
Metric definitions:
Novelty: Evaluate the novelty of the generated scientific hypothesis. The score range should be 0 to 3. 0 means there's no novelty, which indicates that the hypothesis is a paraphrase of the input. 1 means there's slight novelty. 2 means there's moderate novelty. 3 means the hypothesis has strong novelty, which gives new insights beyond the background. Output is an integer.
Relevance: Evaluate the relevance of the generated scientific hypothesis. The score range should be 0 to 3. 0 means there's no relevance. 1 means there's slight relevance. 2 means there's moderate relevance. 3 means they are strongly related. Output is an integer.
Significance: Evaluate the significance of the generated scientific hypothesis. The score range should be 0 to 3. 0 means there's no significance, which indicates that the hypothesis is just a common knowledge. 1 means there's slight significance. 2 means there's moderate significance. 3 means the hypothesis has strong significance, which gives significant insights beyond the background. Output is an integer.

Verifiability : Evaluate the verifiability of the generated scientific hypothesis . The score range should be 0 to 3. 0 means there's no verifiability , which indicates that the hypothesis is not possible to be verified in future work. 1 means there's slight verifiability . 2 means there's moderate verifiability . 3 means the hypothesis has strong verifiability , which means the hypothesis is very likely to be verified in future work. Output is an integer .
User Input : {user input}
Hypothesis: {hypothesis}

By contrast, a *pairwise* evaluator compares two hypotheses at a time, and is asked to say which of them is better according to each of the four criteria. Ties are allowed. A Bradley–Terry model was then fitted to the outputs; see Section C.

### Prompt 9: Pairwise evaluator

You are a senior biomedical reviewer . Compare two hypotheses A and B on four metrics: Novelty, Relevance, Significance , Verifiability .
Instructions :
For each metric , judge and select a winner:
    − "A" if A is clearly superior ,
    − "B" if B is clearly superior ,
    − "0" if they are equal or difference is unclear .
For each, give a concise reason .
Each metric is judged strictly independently .
Novelty: Evaluate the novelty of two scientific hypotheses (A and B) given the user input . For each, assign a novelty score from 0 to 3. 0 means there's no novelty , which indicates that the hypothesis is a paraphrase of the background. 1 means there's slight novelty . 2 means there's moderate novelty . 3 means the hypothesis has strong novelty , which gives new insights beyond the background. Score two hypotheses and compare which one is more novel("A", "B", or "0" if equal or difference is unclear )
Relevance: Evaluate the relevance of two scientific hypotheses (A and B) given the user input . For each, assign a relevance score from 0 to 3. 0 means there's no relevance . 1 means there's slight relevance . 2 means there's moderate relevance . 3 means the hypothesis is strongly related to the background. Score both hypotheses and compare which one is more relevant ("A", "B", or "0" if equal or difference is unclear )
Significance : Evaluate the significance of two scientific hypotheses (H_A and H_B) given the user input. For each, assign a significance score from 0 to 3. 0 means there's no significance , which indicates that the hypothesis is just common knowledge. 1 means there's slight significance . 2 means there's moderate significance . 3 means the hypothesis has strong significance , providing significant insights beyond the background. Score both hypotheses and compare which one is more significant ("A ", "B", or "0" if equal or difference is unclear )
Verifiability : Evaluate the verifiability of two scientific hypotheses (H_A and H_B) given the user input. For each, assign a verifiability score from 0 to 3. 0 means there's no verifiability , which indicates that the hypothesis is not possible to be verified in future work. 1 means there's slight verifiability . 2 means there's moderate verifiability . 3 means the hypothesis has strong verifiability , which means it is very likely to be verified in future work. Score both hypotheses and compare which one is more verifiable ("A", "B", or "0" if equal or difference is unclear )
User Input : {user input}
H_A: {hypothesis_a}
H_B: {hypothesis_b}

## Supplementary References

Ralph Allan Bradley and Milton E. Terry. Rank analysis of incomplete block designs: the method of paired comparisons. *Biometrika*, 39(3-4):324–345, 12 1952. ISSN 0006-3444. doi: 10.1093/biomet/39.3-4.324. URL `https://doi.org/10.1093/biomet/39.3-4.324`.

Paul-Christian Bürkner. Bayesian item response modeling in r with brms and stan. *Journal of statistical software*, 100:1–54, 2021.

Roger R. Davidson. On extending the bradley-terry model to accommodate ties in paired comparison experiments. *Journal of the American Statistical Association*, 65(329):317–328, March 1970. ISSN 1537-274X. doi: 10.1080/01621459.1970.10481082. URL `http://dx.doi.org/10.1080/01621459.1970.10481082`.

D. Firth. Quasi-variances. *Biometrika*, 91(1):65–80, March 2004. ISSN 1464-3510. doi: 10.1093/biomet/91.1.65. URL `http://dx.doi.org/10.1093/biomet/91.1.65`.

David Firth. *qvcalc: Quasi Variances for Factor Effects in Statistical Models*, 2025. URL `https://CRAN.R-project.org/package=qvcalc`. R package version 1.0.4.

Brian K. Kobilka and Robert J. Lefkowitz. G protein-coupled receptors: A century of research and discovery. *Circulation Research*, 134(5):671–693, 2024. doi: 10.1161/CIRCRESAHA.124.323067.

Biqing Qi, Kaiyan Zhang, Kai Tian, Haoxiang Li, Zhang-Ren Chen, Sihang Zeng, Ermo Hua, Hu Jinfang, and Bowen Zhou. Large language models as biomedical hypothesis generators: A comprehensive evaluation. In *First Conference on Language Modeling*, 2024. URL `https://openreview.net/forum?id=q36rpGlG9X`.

David Antony Selby. *Statistical modelling of citation networks, research influence and journal prestige*. PhD thesis, University of Warwick, June 2020. URL `http://webcat.warwick.ac.uk/record=b3690782`.

Jingchen Shao, Jeonghyeon Kwon, Tianpeng Wang, Stefan Günther, Lukas S Tombor, Timothy Warwick, Zaib Shaheryar, Ralf P Brandes, Stefanie Dimmeler, Jan Wenzel, et al. Orphan receptor gpr153 facilitates vascular damage responses by modulating camp levels, yap/taz signaling, and nf-$\kappa$b activation. *Nature Communications*, 16(1):6232, 2025.

Heather Turner and David Firth. Bradley–Terry models in R: The BradleyTerry2 package. *Journal of Statistical Software*, 48(9):1–21, 2012. doi: 10.18637/jss.v048.i09.

