# OpenReview forum: "BioDisco: Multi-agent hypothesis generation with dual-mode evidence, iterative feedback and temporal evaluation"
_TMLR — Withdrawn by Authors_

### Review · Reviewer_oWZE · 2025-12-09

**Summary Of Contributions:**

The paper introduces BioDisco as a multi-agent framework that integrates knowledge graph querying, automated literature retrieval, and an internal critic-driven refinement loop for biomedical hypothesis generation, evaluating it through temporal benchmarks, pairwise model-based comparison, and expert review. The work is ambitious and addresses a genuine gap in hypothesis-generation research, particularly in its emphasis on novelty and significance, the use of dual-mode evidence grounding, and temporal evaluation that restricts information after specified cut-off dates to assess discovery potential.

**Additional Comments:**

I suggest major revision on this paper before it can get considered for publication in TMLR.

**Audience:**

No

**Audience Explanation:**

1. The literature search relies on keyword-based PubMed queries, which may introduce confirmation bias and incomplete evidence retrieval, limiting claims of strong grounding.

2. The evaluation strategy risks circularity because the same family of language models participates in both generating and judging hypotheses, which may inflate performance. The temporal evaluation excludes the “no relation” category, making the task easier and less realistic.

3. Human evaluation is limited to only nine experts in two subfields, which restricts generalizability, and the cardiovascular ratings in particular show variability in perceived novelty.

4. The results also reveal a trade-off: as refinement increases novelty, verifiability tends to decline, meaning the system produces increasingly complex but less testable ideas.

5. Finally, while the system claims potential for genuine scientific discovery, there is no biological validation, so practical impact remains unproven.

**Broader Impact Concerns:**

I do not have any concerns with any broader Impact.

**Claims And Evidence:**

Yes

**Claims Explanation:**

Pros:
1. The temporal setup, with PubMed restricted by publication date and PrimeKG used as a fixed KG snapshot, is a reasonable attempt to test “future discovery,” and the TruthHypo results demonstrate that generated hypotheses often encode the correct polarity of relations.

2. The ablation is also useful conceptually, showing that adding multi-agent structure, tools, and refinement each improves novelty and significance as judged by an LLM, and the Rasch-style analysis of human ratings is a thoughtful way to handle rater and metric effects.

Cons:
However, most of this evidence is weaker than the narrative suggests. The temporal evaluation does not convincingly rule out data leakage from pretraining, and excluding the “no relation” class in TruthHypo artificially simplifies the problem, making the reported precision and recall easier to achieve than in realistic open-ended discovery.

**Requested Changes:**

The submission would benefit from several important revisions to strengthen its claims and improve clarity.

1. First, the temporal evaluation needs more rigorous control of data leakage risks. This could involve demonstrating that hypotheses remain valid even when evaluated on entity pairs or concepts absent from the LLM’s pretraining window, or incorporating explicit leakage checks. The TruthHypo experiment should be expanded to include the “no relation” class, since abstention and negative prediction are essential in real discovery contexts; without this, the current framing overstates the system’s practical reliability. The classifier agent used for mapping predictions should be trained and validated on a substantially larger and independent subset to avoid optimistic bias.

2. The comparison to Biomni and baseline models requires clearer control over tool access, parameter settings, and number of refinement iterations; as written, it remains unclear whether improvements arise from architecture or simply additional evidence retrieval.

3. The reliance on LLM-based judges means evaluation may be self-reinforcing; including human expert scoring for a larger and more diverse pool of hypotheses, or alternative automatic metrics such as citation-supported link prediction, would make the results more convincing.

4.The paper should address the trade-off where refinement reduces verifiability, by introducing a mechanism for uncertainty estimation or abstention to prevent overly speculative hypotheses.

5. Finally, key details should be clarified: the rationale for fixed iteration counts, the cost–benefit balance of evidence retrieval, and precise limits of KG and PubMed coverage.

---

### Review · Reviewer_sN3n · 2025-12-16

**Summary Of Contributions:**

Paper summary

This manuscript proposes BioDisco, a multi-agent framework for the automated generation of biomedical hypotheses. It combines language-model reasoning with dual-mode evidence from knowledge graphs and real-time literature retrieval. Candidate hypotheses are scored along novelty, relevance, significance, and verifiability to guide targeted acquisition of additional evidence and subsequent refinement through an agentic self-critique mechanism. A key contribution is the use of rigorous temporal evaluation to assess whether generated hypotheses can anticipate future scientific discoveries rather than restate existing knowledge. Extensive experiments demonstrate that BioDisco outperforms single-agent baselines and generalist biomedical agents in terms of novelty and scientific significance.

Paper strength

1. The framework uniquely combines structured biomedical knowledge graphs with unstructured, real-time literature retrieval, enabling hypotheses to be grounded in explicit relational structure while remaining informed by the latest scientific findings.

2. The authors provide an open-source Python library that can be executed on standard CPU hardware. This lowers the barrier to adoption and supports broader reuse.

3. The study employs a comprehensive evaluation strategy that combines temporal evaluation for future-discovery prediction with paired comparisons and Bradley–Terry modeling to obtain statistically grounded performance estimates.

Major weaknesses

1. Explicit quantitative comparisons with a broader range of representative hypothesis-generation systems are missing. This makes it difficult to evaluate BioDisco’s gains relative to existing methods fully.

2. The experiments are conducted on datasets of limited size (134 and 300 samples), which may constrain statistical robustness and generalizability of the conclusions.

3. The scoring scheme (novelty, relevance, significance, verifiability) is treated as fixed. The impact and contributions of each score dimension remain unexplored.

4. The use of cosine similarity demonstrates semantic relatedness between generated and gold hypotheses, but does not guarantee logical validity, causal soundness, or biological correctness. High similarity may reflect shared terminology rather than correct mechanistic reasoning. Moreover, the human evaluation is limited in scale and does not explicitly assess consistency or correctness.

5. In the TruthHypo experiments, samples labeled as “no relation” are removed, simplifying the task to binary positive/negative relation prediction. It reduces the evaluation of the system’s ability to refuse to answer when no plausible relationship exists, which may cause hallucinations in real-world use.


Minor weaknesses

1. Section 4 Python Package is more suitable for supplementary material rather than the main content, as it does not directly contribute to methodological or empirical insights.

2. As observed in Figure 6, optimizing for novelty may encourage the addition of unnecessary complex mechanistic details while reducing verifiability. This suggests that the scoring scheme may bias the system toward over-elaboration.

3. All experiments rely exclusively on the GPT-family, leaving the generalizability of the framework unclear. The performance gains can be model-specific behavior rather than architectural advantages.

4. The framework introduces multiple specialized agents, but the necessity and unique contribution of each agent are not fully justified. It remains unclear whether all roles are required or some could be merged without loss of performance.

5. The iterative refinement loop relies on score thresholds and a fixed maximum number of iterations, but the choice of these stopping criteria appears heuristic and is not systematically analyzed.

6. The paper focuses primarily on successful or high-scoring hypotheses, with little discussion of failure modes or low-quality outputs, which is critical for assessing reliability and trustworthiness.

**Audience:**

Yes

**Audience Explanation:**

Researchers studying LLM agents, self-critique mechanisms, and automated hypothesis generation would find this work relevant. The proposed framework illustrates how multiple specialized agents can be orchestrated to generate, refine, and evaluate hypotheses, and how temporal evaluation can be used to assess discovery-oriented claims.

**Broader Impact Concerns:**

No ethical concerns.

**Claims And Evidence:**

No

**Claims Explanation:**

Key claims rely on small-scale datasets, limited baseline comparisons, and semantic similarity metrics that do not guarantee scientific correctness. Consequently, the results are suggestive but not fully convincing in demonstrating the robustness and general applicability of the proposed framework.

**Requested Changes:**

1. Please include additional baselines with explicit quantitative results.

2. Please expand the evaluation to larger or more diverse benchmarks to better quantify the proposed method.

3. Please conduct ablations on scoring components or analyze how sensitive hypothesis selection is to score design.

4. Please include logic or relation-level validation metrics to assess scientific correctness.

5. Please discuss the impact of eliminating “no relation” cases or report a separate analysis to assess hallucination control.

6. Please move Section 4 to the appendix or supplementary materials to improve narrative flow.

7. Please clarify the rationale behind the score design and discuss why such a novelty–verifiability trade-off is expected or acceptable. A more detailed analysis is recommended to interpret the results.

8. Please validate the framework with additional model families or include a discussion on expected generalization.

9. Provide a finer-grained ablation or qualitative analysis showing how specific agents affect hypothesis quality.

10. Please clarify how thresholds were selected, or include a sensitivity analysis showing how performance changes with different iteration limits or stopping rules.

11. Please include representative failure cases or error analysis to assess the model limitations.

---

### Review · Reviewer_5CRL · 2025-12-18

**Summary Of Contributions:**

The paper introduces **BIODISCO**, a multi-agent framework designed to automate the generation of novel biomedical hypotheses. The system utilizes a pipeline of specialized Large Language Model (LLM) agents (Planner, Background, Explorer, Scientist, Critic, Reviewer, Refiner) that collaborate to generate, evaluate, and refine hypotheses.

**Strengths:**
* **Methodological Rigor in Evaluation:** The use of the Bradley-Terry model and Bayesian mixed-effects models for human evaluation is a significant step forward compared to standard mean-score reporting in LLM evaluations.
* **Integration of Tools:** The combination of KG querying (Neo4j) and dynamic literature search addresses the hallucination risks common in "one-shot" generation systems.
* **Ablation Studies:** The paper clearly quantifies the marginal contribution of the multi-agent structure, tool use, and refinement loops.

**Weaknesses:**
* **Trade-off in Verifiability:** The results indicate that while novelty and significance improve with the full system, "verifiability" and "relevance" often decrease or show no significant improvement compared to simpler baselines.
* **Cost:** The full BIODISCO pipeline is significantly more expensive (approx. $0.071 per hypothesis) compared to the baseline ($0.004), raising questions about cost-benefit efficiency.
* **Evaluation Limitations:** The TruthHypo evaluation explicitly excluded negative "no relation" samples, potentially masking the system's ability to reject false positives.

**Audience:**

Yes

**Audience Explanation:**

The paper speaks to several segments of the TMLR community:

1. **LLM and Agent-System Researchers:** The critic–reviewer–refiner loop is a concrete and well-documented example of coordinating multiple agents for complex reasoning tasks.
2. **AI for Scientific Discovery:** The biomedical focus, combined with explicit knowledge-graph integration, addresses a high-impact and actively studied application area.

**Broader Impact Concerns:**

The authors have included a limitations section, but a specific "Broader Impact Statement" regarding the risks of automated science is warranted given the domain.

* The system optimizes for "Novelty" and "Significance" but the evaluation shows a potential degradation in "Verifiability". In a biomedical context, generating highly novel but experimentally unverifiable (or factually hallucinated) hypotheses consumes valuable wet-lab resources and time.
* The authors should briefly address the ethical implication of releasing tools that generate plausible-sounding but potentially erroneous scientific text. They note the lack of a "low-confidence filter"; highlighting the necessity of human-in-the-loop verification before any physical experimentation is crucial to responsible deployment.

**Claims And Evidence:**

Yes

**Claims Explanation:**

The experimental results largely support the authors’ central claim that BIODISCO produces hypotheses that are, on average, more novel and significant than baseline methods.

- **Temporal Evaluation:** Performance on TruthHypo is strong, with high reported precision (e.g., 0.90 for Chemical–Gene pairs) and solid recall. The semantic similarity analysis against Qi et al. further suggests that generated hypotheses align more closely with future ground truth than random baselines.
- **Ablation Results:** The Bradley–Terry analysis (Figure 4) provides clear statistical evidence that both the multi-agent design and tool integration contribute to gains in novelty and significance. Reporting confidence intervals here is especially helpful.
- **Human Evaluation:** Although limited to nine experts, the use of a Bayesian polytomous Rasch model makes the conclusions more credible by accounting for individual rater tendencies.

That said, the evidence around the *refinement* loop is more ambiguous. Refinement does not reliably improve verifiability, a point the authors report candidly. This transparency ultimately strengthens the paper, even if it highlights a real limitation of the approach.

**Requested Changes:**

1. **Explicitly Address the “No Relation” Exclusion**

   The paper notes that the TruthHypo evaluation excludes the “no relation” class, while also acknowledging that abstention is important in real-world settings. This limitation should be discussed explicitly in the *Results* section (e.g., Sections 3.1.2 or 3.1.3), not only later in the discussion. Without the negative class, precision and recall primarily reflect the ability to distinguish between relation *types*, not between true and false hypotheses. A clear caveat near Table 2 would prevent overly optimistic interpretations.

2. **Analyze the Drop in Verifiability**

   Both the ablation study (Figure 4) and human evaluation (Figure 6) show that verifiability often stagnates or declines with the full BIODISCO pipeline. Please include a qualitative analysis—possibly in the appendix—explaining why this happens. Is the Refiner introducing complexity that makes experimental validation harder? Or are highly novel hypotheses inherently more speculative? Discussing two or three concrete examples where refinement reduced verifiability would clarify this trade-off.

3. **Cost–Benefit Discussion**

   Table 7 shows the full system is roughly 18× more expensive than the baseline. A short discussion on whether the observed gains in novelty justify this cost would be valuable. For large-scale hypothesis screening, is the full pipeline realistically deployable?

4. **Additional Detail on Human Evaluation**

   The study relies on nine expert reviewers.  If possible, comment on whether reviewer self-reported confidence correlates with scoring behavior. Even a brief observation would strengthen the human-evaluation section.

5. **Clarify the “Unrelated Gold” Baseline**

   Figure 3 includes an “unrelated gold” baseline. Please state explicitly whether this involves comparing hypotheses from one topic against gold standards from another. This seems implied, but a clear explanation would help readers interpret the similarity floor.

---

### Author Response · Authors · 2026-01-08
**Response to Reviewers**

We sincerely thank Reviewers 5CRL, sN3n, and oWZE for their thoughtful and detailed feedback. We are encouraged by the reviewers recognition of the novelty of the BioDisco framework and the methodological contribution of our evaluation strategy.
Below, we address the major points raised across the reviews and describe how we will incorporate this feedback to further strengthen the manuscript.

**1. The Exclusion of the "No Relation" Class**

We acknowledge the validity of this concern. The original intent of the TruthHypo experiment was to evaluate the system’s generative capacity. Specifically, we focused on its ability to recover valid future scientific relationships rather than its ability to abstain.

We agree that in a practical scientific discovery setting, abstention and the avoidance of false positives are important. Therefore, we reran the TruthHypo experiment with the inclusion of the no relation class. The updated results (below) will be added to the supplementary material and the Results section revised to explicitly discuss this point.
| Group | Class | Precision | Recall | $F_1$ | Support |
| :--- | :--- | :---: | :---: | :---: | :---: |
| Chemical-Gene | negative | 0.617 | 0.580 | 0.598 | 50 |
| | positive | 0.895 | 0.680 | 0.773 | 50 |
| | no_relation | 0.492 | 0.640 | 0.557 | 50 |
| Disease-Gene | inhibit | 0.905 | 0.380 | 0.535 | 50 |
| | stimulate | 0.514 | 0.380 | 0.437 | 50 |
| | no_relation | 0.380 | 0.700 | 0.493 | 50 |
| Gene-Gene | negative | 0.600 | 0.300 | 0.400 | 50 |
| | positive | 0.750 | 0.480 | 0.585 | 50 |
| | no_relation | 0.441 | 0.820 | 0.573 | 50 |

**2. Novelty–Verifiability Trade-off**

The results show that while BioDisco improves Novelty and Significance, it often exhibits a decrease in Verifiability. This may indicate the generation of more complex or speculative hypotheses.

We interpret this trade-off as an interesting scientific result rather than a system weakness: as hypotheses become more novel, such as by proposing multi-step mechanisms or previously unseen entity interactions, they naturally become harder to verify using standard, low-cost experimental assays. We will update the Discussion section to clarify how the inclusion of more mechanistic detail can shift hypotheses from being easily testable to requiring more complex experimental designs. We will also explain why this trade-off is acceptable and expected in the context of ab initio hypothesis generation.

**3. Evaluation Rigor: Baselines, Leakage and LLM-based judges**

We thank the reviewers for raising these important concerns regarding evaluation rigor. Below, we address each point in turn.

* **Data Leakage:** We employed strict temporal cutoffs for all literature retrieval tools. We also ensured that the pretrained weights of the LLMs used in our experiments respect these cutoffs. We believe these measures substantially mitigate the risk of data leakage.
* **LLM as Judge:** We acknowledge the potential risk of circularity when using LLMs for evaluation. However, our human expert evaluation (analyzed using a Bayesian Rasch model) largely corroborates the LLM judge’s findings. This agreement suggests that the LLM judge is a reasonable proxy for this task and enables large-scale evaluation that would be infeasible with human evaluators alone.
* **Baselines:** We compared BioDisco against Biomni, which represents the current state of the art for generalist biomedical agents. Identifying additional open-source and reproducible generative hypothesis systems for direct comparison is challenging, as adapting them for our controlled experiments requires substantial effort (such as employing strict temporal cutoffs). We also note that many recent state-of-the-art systems follow architectures similar to our ablated variants, making these comparisons informative.
* **Cosine similarity:** We acknowledge that cosine similarity captures semantic relatedness and does not guarantee biological correctness. However, many prior hypothesis generation systems rely on similar metrics, such as cosine similarity or BLEU, when comparing generated hypotheses to gold standards. In addition, we performed manual comparisons of generated hypotheses against the gold standard in the supplementary material and found that cosine similarity is a reasonable metric in this setting.

**4. Cost and Efficiency**

The full pipeline is significantly more expensive (~$0.07 per hypothesis) than the baseline. While the relative cost increase is high (18x), we argue that the absolute cost is negligible in the context of biomedical research, where a single wet-lab experiment costs thousands of dollars and human literature review takes hours.

---

### Note · Authors · 2026-01-19

**Comment:**

We thank the reviewers for their considered feedback and have decided to withdraw our submission from TMLR in order to perform additional experiments and revise our manuscript.

**Withdrawal Confirmation:**

I have read and agree with the venue's withdrawal policy on behalf of myself and my co-authors.